# Conditionally Activated (“Caged”) Oligonucleotides

**DOI:** 10.3390/molecules26051481

**Published:** 2021-03-09

**Authors:** Linlin Yang, Ivan J. Dmochowski

**Affiliations:** Department of Chemistry, University of Pennsylvania, Philadelphia, PA 19104-6323, USA; linliny@sas.upenn.edu

**Keywords:** caged oligonucleotides, transcriptome in vivo analysis, enzyme activation

## Abstract

Conditionally activated (“caged”) oligonucleotides provide useful spatiotemporal control for studying dynamic biological processes, e.g., regulating in vivo gene expression or probing specific oligonucleotide targets. This review summarizes recent advances in caging strategies, which involve different stimuli in the activation step. Oligo cyclization is a particularly attractive caging strategy, which simplifies the probe design and affords oligo stabilization. Our laboratory developed an efficient synthesis for circular caged oligos, and a circular caged antisense DNA oligo was successfully applied in gene regulation. A second technology is Transcriptome In Vivo Analysis (TIVA), where caged oligos enable mRNA isolation from single cells in living tissue. We highlight our development of TIVA probes with improved caging stability. Finally, we illustrate the first protease-activated oligo probe, which was designed for caspase-3. This expands the toolkit for investigating the transcriptome under a specific physiologic condition (e.g., apoptosis), particularly in specimens where light activation is impractical.

## 1. Introduction

Synthetic oligonucleotides (oligos)—typically short DNA or RNA strands—have revolutionized many areas of biomolecular research and clinical practice. These synthetic biopolymers often have the same chemical structure as naturally produced forms and achieve their function through base pairing with complementary strands, e.g., 18–25 nucleotide PCR primers, antisense oligos, small interfering RNAs (siRNAs), microRNAs (miRNAs), ribozymes, and CRISPR single-guide RNAs (sg-RNAs). Synthetic modifications of oligos to increase binding affinity, specificity, or nuclease stability are also common, e.g., phosphorothioation or other backbone modifications including peptide and morpholine units, as well as base modifications, e.g., 2′-alkoxy and fluoro RNA. Longer (typically 20–60 bases) single-stranded synthetic RNA and DNA aptamers that use base-pairing to adopt compact 3-dimensional structures and are similar in concept to antibodies, have been widely investigated for targeting small molecules, proteins, and even cell surfaces. The versatility and ease of automated solid-phase oligo synthesis using phosphoramidite chemistry developed in the late 1970s makes it possible to program oligo sequence and incorporate other functional moieties in a straightforward manner [1,2].

Conditionally activated (“caged”) oligos are designed with their biological function temporarily blocked until restored by programmed stimuli. Development of caged oligos over the past several decades has provided spatiotemporal control required to study dynamic biological processes in complex microenvironments. As revealed in recent studies in cancer research and neuroscience, tools capable of capturing spatiotemporal patterns have provided insights for understanding cell development [3], disease progression [4], and environmental influence [5].

Some recent reviews have provided perspectives on oligo pro-drugs [6], and optochemical tools developed with oligos [7]. In this work, we reviewed different designs of caged oligos and applications in biological environments. Following on our earlier reviews [8,9,10], we present recent efforts from our laboratory focusing on the advancement in caging strategies including cyclization and enzyme activation, as well as improvements in caged oligo synthesis. We also update a series of oligo probes applied for single cell transcriptome analysis. 

## 2. Methods for Blocking the Function of Oligos

Oligo function relies on direct base pairing to one or more complementary oligos, or internal base-pairing to form secondary or tertiary structures required to bind protein targets. Thus, a central strategy for caging oligos to modulate their biological function is to inhibit base pairing. Over the past several decades, four major caging strategies have been explored in the literature and each comes with its own benefits and challenges (Figure 1).

### 2.1. Incorporating a Caging Group

The most common caging strategy is to introduce steric hindrance with a bulky caging group attached to the oligo (Figure 1A) [11,12]. Various caging moieties have been installed on nucleobases to interrupt normal Watson-Crick base pairing. The caged nucleosides are generated in the form of phosphoramidites for the ease of incorporation through solid-phase synthesis at pre-selected sites. Generally, caging groups replace hydrogen on exocyclic heteroatoms, e.g., N^4^ of cytidine, O^4^ of thymidine or uridine, N^6^ of adenosine, and O^6^ of guanosine [13,14]. N^3^-protected thymidine was also developed for biological applications [15]. Less commonly, Anhäuser et al. introduced caging groups to purine imine positions. Aryl ketones such as benzophenone and α-hydroxyalkyl ketone were chemically or enzymatically installed at the N^7^ position of guanosine or the N^1^ position of adenosine, which successfully blocked the binding of translation initiation factor eIF4E [16]. Zhang et al. applied tRNA guanine transglycosylase (TGT) to introduce a bulky pre-queuosine1 (preQ1) derivative, biotin-Bac-PreQ1, which suppressed translation of mRNA [17]. The phosphodiester linkage also provides an effective caging position. In earlier examples, translation of mRNA [18] or transcription of plasmid DNA [19] was suppressed via “statistical labeling” with caging groups non-specifically placed on multiple phosphate groups. Shah et al. introduced the earliest version of caged siRNA by nonspecifically attaching photolabile 4,5-dimethoxy-2-nitrophenylethyl (DMNPE) groups to phosphate groups [20]. The 2′ position on ribose serves as another promising caging site. MacMillan et al. first introduced a 2′-hydroxyl caged adenosine phosphoramidite specifically to the cutting site of an RNA substrate and prevented recognition and cleavage by the hammerhead ribozyme [21]. Recently, the Kool laboratory installed a series of alkyl imidazole reagents to nucleophilic 2′-hydroxyl groups, post-synthetically under superstoichiometric conditions. Acylation disrupted proper folding, which achieved structural-functional control of fluorescent RNA aptamers [22,23,24]. 

In many caged oligo applications, it is non-trivial to identify the key “interaction sites” that can give maximum disruption with a single caging group. Even with a priori knowledge of the “interaction sites”, nonspecific labeling strategies usually require multiple oligo modifications in order to block biological function sufficiently, which in turn poses challenges for restoring binding affinity after activation [25]. 

The Tang laboratory developed a series of caged siRNAs with a single modification at the 5′ end of a siRNA antisense strand, a position considered critical for the formation of RNA-induced silencing complex (RISC) by disrupting the binding of Ago2 protein. A single bulky group such as vitamin E [26], aptamers [27], cyclic RGD peptides [28], folic acid [29], cholesterol [30], and dextran [31] was attached through a 2-nitrophenyl photocleavable linker, which interrupted the RNAi protein machinery until illumination released the protecting group.

Very recently, an unique macrocyclic protecting group strategy was introduced by Acevedo-Jake et al. Target binding of oligo was blocked as a rotaxane with a macrocycle encircling the oligo axle [32]. A rotaxane was formed during the Cu(I)-mediated alkyne-azide cycloaddition (CuAAC) reaction between two DNA strands modified with azide and alkyne at each end. Cu(I) coordinated to the bipyridine-containing macrocycle directed the reaction to proceed within the macrocycle cavity. While showing the potential of using macrocycles as an effective caging group, the study did not present an activation method to restore oligo function.

### 2.2. Binding to a Complementary Blocking Strand

Another widely used caging approach is to prehybridize the biologically active oligo with a complementary blocking strand (Figure 1B). Subsequently, decreasing the binding affinity of the blocking strand can free the functional strand for binding to the actual target. A simple model is to use a long linear blocking strand with one or multiple stimuli-cleavable linker(s) inserted. Once activated, the blocking strand should break into smaller pieces that release from the functional oligo [33,34,35]. Another method is to use a cleavable linker that covalently attaches the functional strand to the blocking strand to form a stable hairpin structure. If designed correctly, cleavage of the linker should significantly decrease the affinity of the two complementary strands [36,37,38,39]. In this blocking-strand strategy, activation generates a structural change that can be easily monitored by Förster resonance energy transfer (FRET). This method takes into account the binding equilibrium between the functional strand and three partners-blocking strand, broken blocking strands (if multiple linkers are incorporated), and target. It is critical to select properly the position to install a blocking strand and the correct length for both the functional and blocking strands. The potential biological activity of the released blocking strand(s) is also important to consider.

Jain et al. applied this caging strategy in a CRISPR-Cas9 system and introduced the first photocaged sgRNA with a complementary single-strand DNA containing multiple nitrobenzyl photocleavable linkers. This blocking strand significantly reduced editing capability of the CRISPR system on the GFP mRNA target until photoactivation [40]. Tan et al. developed caged aptamers with a 10-base complementary strand connected by a photocleavable linker and triethylene glycol (TEG) spacer forming a hairpin structure. The stable hairpin structure prevented target binding until the blocking strand was released [41]. The Lu laboratory presented a caged ATP-sensing aptamer using a longer (20-base) blocking strand also with a 2-nitrobenzyl photocleavable linker inserted in the middle. A fluorophore was placed at the 5′ end of the aptamer and a quencher on the opposing 3′ end of the blocking strand. The caged aptamer was delivered into mitochondria in living HeLa cells. Fluorescence was only observed after light activation when the aptamer was freed to bind ATP and adopt a new conformation that displaced the quencher. The sensor successfully detected ATP concentration fluctuations caused by Ca^2+^ or oligomycin treatment [42].

### 2.3. Segmentation

An oligo loses binding affinity when cut into pieces and regains function when ligated to form the full-length oligo (Figure 1C). The Abe group introduced a “build-up” approach by delivering inactive short oligo segments which were reassembled to the active full-length form upon activation. In their first example, a 25-base siRNA was segmented into 19- and 6-base pieces for the sense strand, and 18- and 7-base pieces for the antisense strand. At the segmenting point of each strand, the two ends were separately modified with an iodoacetyl (IAc) group and phenyl disulfide-substituted phosphorothioate (PS) group. Upon cellular uptake, disulfide bond was cleaved by intracellular glutathione (GSH) and automatically ligated with IAc group to form the active species [43]. In this example, it is beneficial to deliver shorter strands for in vivo study to reduce immune response. However, the ligated strand may not have the function fully recovered when a different chemical structure is introduced and phosphodiester backbone is interrupted. More recently, Kimura et al. combined the build-up approach with a circular caging strategy to make caged siRNA [44]. The two strands of siRNA were cyclized containing 2-nitrobenzyl photocleavable linker or a disulfide bond, which could be linearized by light irradiation or reduction by intracellular GSH, and subsequently formed a functional siRNA double strand.

### 2.4. Cyclization

A fourth caging strategy is to enforce a secondary structure that masks oligo structure and function. For example, a short circular oligo is less capable of binding to a complementary target due to its curvature (Figure 1D). Cyclization with a cleavable linker can cage an oligo, until activation linearizes the oligo to restore its binding affinity. Our laboratory first introduced a circular caged oligo in 2011 [45]. A 33-base DNAzyme was synthesized using standard phosphoramidite chemistry with a nitrobenzyl photocleavable linker included. 5′ Phosphate and 3′ hydroxyl groups were connected by the enzyme Circligase in 40% yield. The circular DNAzyme showed a 10-fold decrease in activity compared to the linearized version. Tang et al. chemically cyclized antisense DNAs of different sizes by introducing carboxylic acid and amine groups at the ends [46]. More recently, Patrick Seyfried et al. synthesized bis- and tris-azido linkers which could tether alkyne-modified thymine. The size of the circular oligo, which is critical to the caging effect, can be adjusted from the position of thymine. For longer oligos, biscyclization can be introduced by the tris-azido linker, which effectively caged a 95-base aptamer [47]. Yamazoe et al. synthesized a dimethoxynitrobenzyl (DMNB)-based bifunctional photocleavable linker with an N-hydroxysuccinimide ester and a chloroacetamide group to cyclize a 25-nucleobase morpholino oligo (MO). Zebrafish injected with circular MO showed no morphological changes without light activation [48]. Similarly, the Dore laboratory developed a circular caged MO to control the expression of glutamic acid decarboxylase (GAD) and study γ-amino butyric acid (GABA) mediated signaling in zebrafish. A caged MO avoided immediate knockdown of target *gad1* gene which would result in lethal craniofacial defects [49]. An improvement in the synthesis involved replacing the adipoyl amide linking moiety with a simple propargyl unit. This change reduced the linker synthesis to 5 steps, with a 5.6-fold increase in total yield to 28% and the final product was bench stable and did not require −30 °C storage under an inert atmosphere [50]. Zhang et al. introduced circular siRNAs which substantially reduced off-target effects [51]. Compared to the other three oligo caging methods, circularization is the most atom-efficient approach as no additional synthesis involving blocking groups or complementary strands is needed. However, synthesis of circular oligos in high yield can be challenging. Depending on the sequence and backbone modification, optimization of the size of the circle may also be necessary.

## 3. Activation Stimuli

Functional inhibition is only the first step of caging. It is equally important to be able to restore function under control of a stimulus. Various external and internal stimuli have been employed with corresponding caging moieties to achieve different applications.

### 3.1. Light Activation

In 1978, Hoffman and co-workers caged ATP with a 2-nitrobenzyl moiety placed on the γ-phosphate group, and studied the function of the Na/K pump (which is powered by ATP) [52]. Since then, many photo-caging groups have been developed for oligos with multiple applications in gene expression regulation [8,9,10]. O-nitrobenzyl derivatives are some of the most commonly used caging groups [53,54,55]. Light irradiation (365–405 nm) triggers the transfer of the benzylic proton to the nitro group forming an *aci*-nitro intermediate, which leads to the cyclization between the nitronic acid and the benzylic carbon. Decomposition of the heterocycle yields the desired phosphoester. Ordoukhanian and Taylor first developed a phosphoramidite with a nitrobenzyl cleavable linker [56], which enabled convenient incorporation in blocking strand oligo synthesis. Coumarin derivatives are another popular family of caging groups, with some variants showing superior two-photon absorption properties. The Heckel laboratory introduced a 7-diethylaminocoumarin cleavable linker as phosphoramidite, which was incorporated into the loop region of hairpin DNA [57]. Two-photon irradiation at 780 nm activated DNA immobilized in a hydrogel, showing greater 3D resolution and depth penetration compared to one-photon activation. To employ photoactivation at longer wavelengths, Li and co-workers packaged caged oligo onto lanthanide-doped upconversion nanoparticles (UCNPs), which converted near-IR 980 nm light into 365 nm emission. Multiple sensors, including ATP-sensing aptamer [58], light-activated DNA i-motif for pH sensing [59], miRNA probe [60], and Zn^2+^-specific DNAzyme sensor [61] were all successfully applied in vivo under the control of near-IR light. 

Ruthenium (Ru) bipyridyl complexes provide a versatile molecular photo-caging strategy. Studies have shown that [Ru(bipyridine)_2_L_2_]^2+^ complexes can undergo ligand (L) exchange with solvent upon irradiation with 400–500 nm (1-photon) or 800–1000 nm (2-photon) excitation [62]. Our laboratory developed the first Ru photocleavable linker, Ru^2+^(bipyridine)_2_((3-ethynylpyridine)_2_) (RuBEP), which was incorporated into circular caged morpholinos (MOs). MOs targeting the zebrafish genes *chordin* or *no tail* achieved gene knockdown upon irradiation with 450 nm light [63]. Subsequently, additional Ru photoactive crosslinkers were developed. Ru(bipyridine)_2_(3-pyridinaldehyde)_2_ (RuAldehyde), a compound related to RuBEP, could also undergo rapid ligand exchange with irradiation at visible wavelengths (400–500 nm) [64]. And, by replacing one or both bipyridines with biquinoline ligands, a series of sequentially red-shifted crosslinkers was synthesized and found to be suitable for multiplexing applications [65].

Using near-UV, visible, or near-IR light as the external stimulus has many advantages as it introduces few side reactions, allows convenient modulation with high spatiotemporal resolution, and is orthogonal to most natural biological processes. An external light trigger can be easily applied and removed noninvasively, and directed at a precise time to a specific location with tunable intensity. 

Korman et al. combined photoactivation with single-molecule fluorescence detection to resolve the folding pathway of a self-cleavable twister ribozyme of *Oryza sativa* [66]. Conventionally, folding coupled with self-cleavage was difficult to study due to rapid conversion after the active conformation was formed. In this work, a single caged guanosine modified with a *p*-hydroxyphenacyl moiety enabled high spatial (~75 µm) and temporal (≤30 ms) control with 405 nm light irradiation. Light activation has also been applied in multiple ways to the CRISPR-Cas9 gene editing system. Zhou et al. caged the sgRNA with four 6-nitropiperonyloxymethylene (NPOM) modified nucleotides (uridine or guanosine) evenly distributed in the 20-base target spacer region. The system successfully achieved light-activated gene editing in both mammalian cells and zebrafish embryos targeting exogenously supplied plasmid or the endogenous genome [67]. Similarly, Liu et al. developed a caged sgRNA by replacing two or three uracils in the spacer region with NPOM-modified deoxynucleotide thymine. The system achieved by far the highest spatiotemporal resolution for the CRISPR system, at the submicrometer and second scales. Tested with four endogenous loci in HEK 293 cells, only negligible insertions or deletions (indels) were detected without light activation and up to 52% double-strand breaking was achieved with 30 s irradiation at 365 nm. The precise control enabled observation of repair dynamics after double-strand cleavage [68].

### 3.2. Cellular Oxidation and Reduction (Redox) Activation

The intracellular environment contains a high concentration of reactive oxygen species (ROS), reactive nitrogen species (RNS), and GSH, which altogether contribute to redox homeostasis. A well balanced redox environment is critical to cell survival. Levels of some redox species are considered good biomarkers in disease progression. Activation by redox species makes oligos only function in a specific intracellular environment. Mori et al. designed a series of caged nucleosides reactive to H_2_O_2_, which is at elevated concentration in cancer cells [69]. Pinacol arylborane groups were introduced to the bases to sterically block duplex formation. However, due to the hygroscopicity of arylboronic acid moieties, direct incorporation was difficult via solid-phase synthesis in an anhydrous environment. In this work, only boronated dT was converted to phosphoramidite, and a short antisense oligo was caged with two or three boronated dTs. The knockdown effect was only observed after treatment with 10 μM H_2_O_2_. The Urata group introduced 2′-O-methyldithiomethyl (MDTM)-modified RNA and the protecting group could be removed in the intracellular reducing environment [70]. siRNAs with 2′-O-MDTM-modified antisense strand were deprotected in 1 h in the presence of 10 mM GSH in vitro, and reduced target gene expression more effectively compared to non-modified siRNA in cell studies [71,72]. The same laboratory also introduced a cyclic disulfide *trans*-5-alkyl-1,2-dithiane-4-yl moiety to the phosphodiester linkage. With 10 mM GSH, the cyclic disulfide group was almost completely removed in 75 h [73]. Ikeda et al. synthesized a deoxyguanine phosphoramidite with 4-nitrobenzyl (NB) group at O^6^ position (dGNB), which could be deprotected in a reducing environment with Na_2_S_2_O_4_ or nitroreductase. dGNB was incorporated into the 5′ end of thrombin-binding DNA aptamer and G-quadruplex structure only formed after treatment with reductants [74,75]. Notably, the main motivation for developing redox-responsive oligos is to increase nuclease resistance and assist intracellular delivery. Although it has been shown that reducing environment is an effective way to convert a protected oligo to its functional form, more thorough characterization of the caging effects is still needed. 

### 3.3. Enzyme Activation

Enzymes play fundamental roles in many biological processes. Enzymatic activation could connect oligo function to specific biological or biochemical conditions. However, until now, few enzyme-activated oligos have been introduced. Most prior studies harnessed reductase activity, which is overexpressed in solid tumors under hypoxic conditions. The Ono group has developed multiple reductase-responsive linkers for oligos [76,77]. In one example, 3-(2-nitrophenyl)propyl (NPP) group was pre-attached to the phosphodiester linker and phosphoramidite was synthesized. With multiple modifications, DNA oligos showed a lower melting temperature (Tm) with a complementary strand, stronger resistance to both exonucleases and endonucleases, and higher cellular uptake compared to an unmodified control. The Chen lab presented the first in vivo application of enzyme-activated morpholino antisense oligos sensitive to *Escherichia coli* nitroreductase NfsB [78]. In this work, a 4-nitrobenzyl (4-NB) bifunctional linker was first synthesized to connect the two ends of *n*-hydroxysuccinimide ester and chloroacetamide on the MO. An in vitro study showed that the circular MO could be completely linearized by 4 nM NfsB enzyme within 1 h, with a response that varied in a dose- and time-dependent manner. A circular 25-base MO targeting mesodermal T-box transcription factor *no tail-a* (*ntla*) was introduced into zebrafish, where a phenotypic change was only observed when co-injected with NfsB mRNA. 

Meade et al. took enzyme activation in a different direction by making ribonucleic neutrals (RNNs) that incorporated S-acyl-2-thioethyl (SATE) phosphotriester group on the phosphate backbone [79]. The backbone could be converted into the native form by intracellular thioesterases. siRNN with multiple phosphotriester groups showed RNAi responses in mice with strong serum stability and absence of an immune response. Although not designed for use as caged oligos, the authors observed disruption of double-stranded siRNA formation from excess SATE modification.

### 3.4. Heat Activation

Temperature plays a significant role in many chemical reactions, however there is not much room for large temperature changes for in vivo applications. Heat activation is more often seen as a deprotection method used for solid-phase synthesis. The Beaucage laboratory introduced multiple thermolytic groups onto phosphoramidite phosphate groups for rapid deprotection and to avoid side products after solid-phase synthesis [80]. In an earlier example, phosphoramidites including 2-(*N*-formyl-*N*-methyl)aminoethyl (fma) thiophosphate protecting group were developed and applied to CpG-containing DNAs [81]. In vivo studies with mice showed the modified CpG drug had a comparable efficacy but a delayed response. Later, a series of heat-sensitive thiophosphate protecting groups was screened, which expanded the window of deprotection half-time from 17 min to 93 d [82]. Madaoui et al. also included thermolytic groups methylthioethanol and 2,2′-thiodiethanol into cyanoethyl phosphoramidites and solid support, providing a new way to synthesize 3′- or 5′-phosphate or thiophosphate oligos [83]. Most recently, Heemstra and coworkers introduced a versatile glyoxal caging strategy for thermolytic control of RNA and DNA structure and function [84].

## 4. Recent Applications of Caged Oligos

### 4.1. Circular Light-Activated Antisense DNA

As discussed earlier, cyclization provides unique benefits as an oligo caging strategy. However, syntheses described in the literature are generally laborious with low yields [45,47,50]. Our laboratory has introduced an efficient method to synthesize circular caged oligos through intramolecular copper(I)-catalyzed azide-alkyne cycloaddition (CuAAC) (Figure 2A) [85]. A linear strand including all nucleobases and a nitrobenzyl photocleavable linker was first synthesized on solid phase starting with 3′-alkyne CPG and ending with a 5′-amine. The 5′ end was later converted to azide by reacting with azidobutyrate N-hydroxysuccinimide. Cyclization was conducted at rt in 1 h with near quantitative yield. The product, after running through a NAP-5 desalting column, did not require HPLC purification. A short stem was included to bring the ends closer to assist the reaction and further stabilize the structure. The Cy3-Cy5 FRET pair at the ends of the stem indicated caging and activation. In the caged form when Cy3 and Cy5 are placed close together, excitation of Cy3 will result in Cy5 fluorescence from energy transfer. After photolysis, the caged oligo is linearized to restore binding to the complementary strand. Meanwhile, the Cy3 signal increases and Cy5 signal decreases as the distance between the two fluorophores increases. This synthesis strategy was successfully tested on different sequences with different stem lengths. A circular antisense DNA (AS-ODN) was synthesized targeting *gfap*, a marker gene encoding glial fibrillary acidic protein (GFAP) in astrocytes. Caged oligo was delivered into cells with the assistance of a noncovalently attached cell penetrating peptide, PepFect-6 (PF6). An immunofluorescence assay demonstrated a 10-fold decrease of GFAP protein expression in cells treated with caged AS-ODN and light irradiation. Cells loaded with AS-ODN but not receiving photoactivation remained at a comparably high protein expression level as blank sample (Figure 2B). This result suggested that the circular caged oligo remained stable in cells and was able to regulate gene expression upon light irradiation.

### 4.2. Transcriptome In Vivo Analysis (TIVA)

Single cell transcriptome analysis has drawn increasing interest as a method for studying cell heterogeneity within live tissue, making it possible to identify rare cell populations [86] and gene regulatory networks [87], or reconstruct cell lineage [88]. Our laboratory developed a multifunctional caged oligo for single-cell Transcriptome In Vivo Analysis (TIVA), which enables mRNA isolation from selected single cells in living tissue. The initial design [89], “18/7/7”, featured a hairpin structure with an 18mer 2′-F poly-U capture strand and two 7mer 2′-OMe poly-A blocking strands connected by two 2-nitrobenzyl photocleavable linkers (Figure 3A). Upon irradiation with 405 nm laser, the two blocking strands are released, and the capture strand binds to the poly-A tail of mRNA. A Cy3-Cy5 FRET pair was installed at the hairpin termini to monitor cellular uptake and activation. Disulfide-linked (D-Arg)_9_ peptide was attached at the 5′ end to promote cellular uptake, which is removed in the intracellular reducing environment. A biotin handle on the 3′ end of the capture strand allowed mRNA extraction with streptavidin beads. TIVA probe was applied in rodent and human brain tissue revealing different expression patterns (and greater heterogeneity) in single neurons compared to the bulk average. To extend applications in whole, living organisms, which generally require longer time courses in nuclease-abundant environments, “22/9/9 (GC)” TIVA probe was introduced with multiple modifications [90]. The capture and blocking strands were lengthened from 18mer and 7mer to 22mer and 9mer, providing greater pre-photolysis caging stability and post-photolysis mRNA binding affinity. A terminal GC pair favored proper alignment of the duplex to avoid a potential sticky end coming from nonspecific A-U binding in the original version (Figure 3B). Finally, a phosphorothioated backbone showed significant nuclease stability during 48 h time course in the presence of 10% serum. More recently, we introduced a third-generation TIVA probe, “22/12/8 (GC)_2_” [91], with an additional GC pair, a flexible TEG linker at the turn and length adjustments for the blocking strands (Figure 3C). These modifications further improved duplex alignment, providing even more robust caging integrity. mRNA isolation experiments from murine brain tissue exhibited negligible pull-down prior to photolysis and significant pull-down after photoactivation (Figure 3D).

### 4.3. Caspase-Activated Probe

While light activation has advantages of easy control and rapid response, this application is also limited to within thin tissue or optically transparent model organisms. Our laboratory has developed the first protease-activated oligo probe reporting on the activity of caspase-3, an enzyme activated during apoptosis [92]. The oligo probe has a similar hairpin structure as TIVA with a GC pair aligning the ends and FRET pair indicating the caging state. A poly-U 2-F′ RNA was connected to a poly-A peptide nucleic acid (PNA) blocking strand by a DEVDK(N_3_) peptide linker instead of a photocleavable linker (Figure 4A). PNA is an oligo analog with nucleobases attached to a polyamide backbone. In this design, the PNA oligomer and peptide linker were synthesized contiguously on the solid phase synthesizer, which overcame the charge repulsion between RNA and a negatively charged peptide. The PNA-peptide blocking strand was conjugated to an alkyne-modified RNA strand through CuAAC reaction in ~30% yield. The caged probe was delivered into HeLa cells by Pep-3, a noncovalently attached CPP. While maintaining high FRET signal in control cells, FRET efficiency decreased from 66.4% to 37.6% in apoptotic cells triggered by staurosporine (STS). We considered that an enzyme-activated probe could be a complementary tool in TIVA applications for studying transcriptomic changes under specific physiological conditions. However, full-length RNA isolation was particularly challenging during apoptosis as degradation by ribonucleases happened on a similar time scale as upregulation of caspase-3. Improvements in the kinetics of oligo activation/peptide proteolysis would likely improve on the ability to capture intact mRNAs. We note that the probe design can also be readily extended to other protease substrates.

## 5. Summary

The term “caged” was first introduced with light-activated ATP by Hoffman in 1978, which described a bioactive molecule whose function was temporarily masked by a protecting group that could be removed with light. After more than four decades of research, caging has evolved into a more encompassing biochemical concept. With improved synthetic methods, and multiple function-blocking strategies accompanied by different activation stimuli, a variety of caged oligos have been developed to answer specific biological questions. Admittedly, many technologies remain at the proof-of-concept stage, however, multiple tools have empowered basic research and driven new discoveries, and these technologies are continuing to advance. There is great potential to develop more accessible synthetic methods for incorporating robust caging stability/function blocking and rapid activation response in biologically relevant environments. An expanding array of commercially available phosphoramidites and simple post-synthetic modifications will continue to drive adoption from biologists. As clinical studies on gene-based therapies and oligo vaccines are making rapid progress, spatiotemporal control from caged oligos may bring benefits to improve pharmacokinetics and reduce cytotoxicity. At the same time, other activation methods as valuable complements to photoactivation are worth more attention.

## Figures and Tables

**Figure 1 molecules-26-01481-f001:**
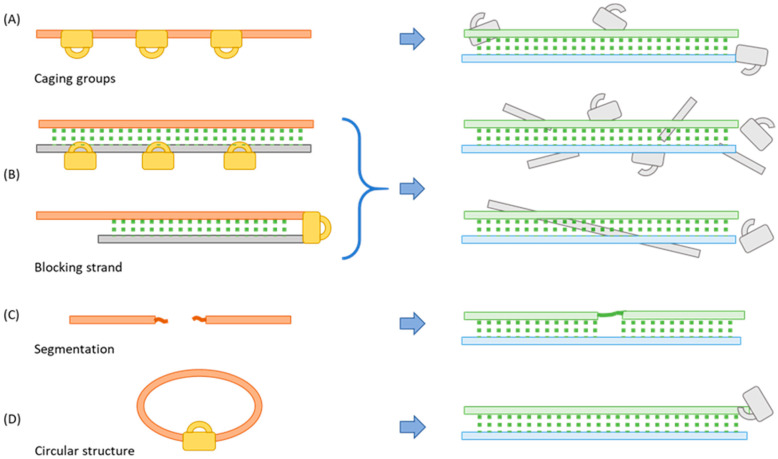
Four oligo caging strategies described in the literature: orange bars on the left illustrate the caged oligos while the green ones on the right are activated, and blue bars represent the target strands. (**A**) Incorporate bulky caging groups which are removed during activation; (**B**) Prehybridize with a complementary strand (the grey bar) that incorporates one or more cleavable linkers, which upon activation allows target binding; (**C**) Break the oligo into small segments and ligate the segments during activation; (**D**) Enforce a circular structure with a cleavable linker; activation linearizes the oligo, restoring its binding affinity.

**Figure 2 molecules-26-01481-f002:**
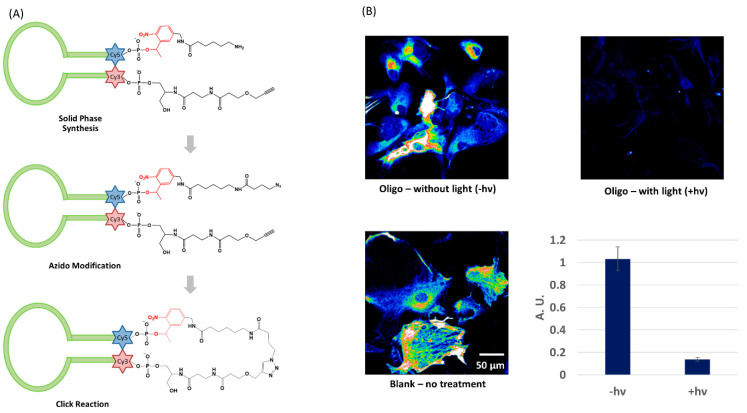
Circular caged oligo for gene expression regulation: (**A**) 3-step synthesis to make circular caged oligo; (**B**) Immunofluorescence for selectively imaging GFAP for samples loaded with AS-ODN, with or without light irradiation, and blank sample without any treatment. Quantification was conducted with average pixel intensity in individual cells normalized to the blank sample. Adapted from ref [85]. The figure is adapted with permission from: Yang, L. et al. *ChemBioChem* 2018, 19, 1250–1254. Copyright 2018 Wiley-VCH Verlag GmbH & Co. KGaA, Weinheim.

**Figure 3 molecules-26-01481-f003:**
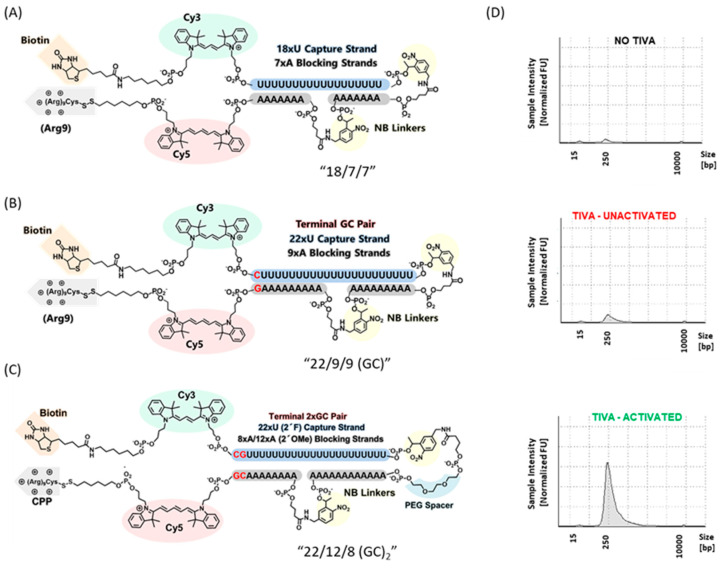
Three generations of TIVA probes for mRNA isolation: structure of (**A**) “18/7/7”; (**B**) “22/9/9 (GC)”; (**C**) “22/12/8 (GC)_2_”; (**D**) Bioanalyzer traces of amplified RNA (aRNA) made from controls and RNA isolated using 22/12/8 (GC)_2_ TIVA from samples that had no TIVA loaded, or loaded with TIVA, without or with light activation. Adapted from ref [91]. The figure is adapted with permission from Yeldell, S. et al. *ACS Chem Biol.* 2020, 15, 10, 2714–2721. Copyright (2020) American Chemical Society.

**Figure 4 molecules-26-01481-f004:**
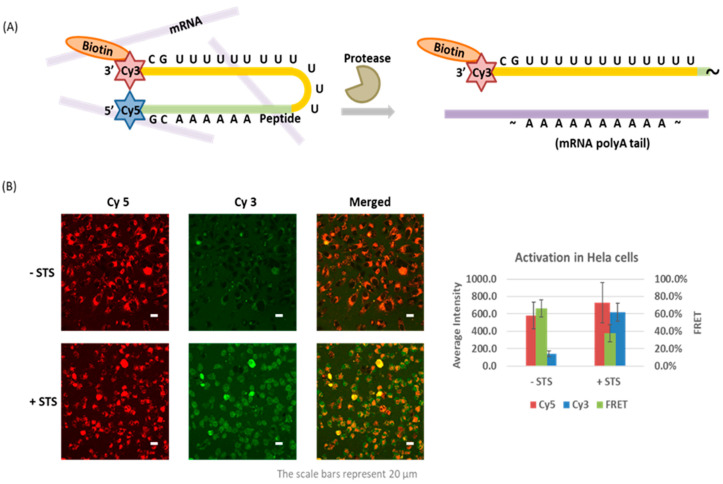
Protease-activated oligo probe: (**A**) Caged probe features a hairpin structure including peptide linker, FRET pair and biotin handle. Caspase-3 cleaves the peptide substrate, promotes release of capture strand (yellow) and binding of mRNA target (purple); (**B**) Confocal fluorescence micrographs and quantification of caged probe remaining high FRET (high intensity of Cy5 fluorescence) in control sample and low FRET (high intensity of Cy3 fluorescence) in apoptotic cells, 8 h after staurosporine treatment. Adapted from ref [92]. The figure is adapted with permission from Yang, L. et al. *Bioconjug. Chem.* 2020, 31, 9, 2172–2178. Copyright (2020) American Chemical Society.

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
