# Peer review of "Conditionally Activated (“Caged”) Oligonucleotides"

_molecules, 2021, doi:10.3390/molecules26051481_

Round 1

Reviewer 1 Report

The authors are internationally recognized experts in the field of chemical synthesis and use of "caged" oligonucleotides : these are sort of "pro-drugs" to be specifically activated by various stimuli (heat, light, chemical reduction, enzyme activity), adapted to various cell cultures or animal models to be studied.  The present manuscript is an update of a previous review from the same research group published in 2015 (ref 50). 

The manuscript is excellent, very clear and well organized , starting with a general overview of the field and providing more details on the most recent technologies (mostly contributed by the group) i.e. antisense oligonucleotide cyclization, TIVA that allows for mRNA isolation from single cells, and protease-activated oligonucleotide probes.

Comments:

In the introduction could the authors refer to the previous review from their group (ref 50) and to more recent reviews from other groups (such as Vasseur 2018, Chen 2019 ...), and explain the novelty of their contribution in the present manuscript?

Line 22: please define oligonucleotide as "oligo" here and keep using this abbreviation all over the text (including line 139)

Line 26: could you be more accurate about CRISPR primers and mention small guide RNAs, thus introducing the sgRNA abbreviation that will be used line 110

Lines 49-50: please clarify/correct English for the legends to Fig1A and B

Line 65: avoid redundancy : mRNA transcripts => mRNAs

Lines 82 and 85: could you be more precise in each case by defining the step at which interference with RISC activity occurs

Line 88: please clarify/correct English

Line 100: motif => method, strategy?

Line 111:  "editing function on GFP" is laboratory slang, confusing protein and DNA molecules! Could you better explain the experimental read out of this study?

Lines 112, 126, 127, 151, 153: please standardize numbering in the manuscript, either nucleotides (nt) or bases

Line 123: please clarify/correct English

Line 130: please mention it's the reduced form of gluthation, and define GSH that will be used line 137

Line 149: => The circle size

Line 163: please clarify "atom-efficient approach"

Line 171: could you add a sentence to explain the mechanism behind light activation of a caged oligonucleotide?

Line 184 and 198: please standardize NIR => near-IR (similar to near-UV on line 198)

Line 187: define (bpy) here

Lines 201-202: avoid redundancy :tunable, fine tuning

Line 204: self cleavable twister ribozyme of Oryza sativa

Line 212: "transfection" refers to cells receiving foreign DNA, e.g. cells are transfected with/by a plasmid, and thus "transfected plasmid" is not correct.

Line 305, legend to Fig 2 B: please explain the different colors observed in the figure, resulting from immunofluorescence  (instead of immunohistochemistry as stated) staining of GFAP , and use of a secondary antibody coupled to Alexafluor 647 (based on methods in original publication), DAPI for DNA staining (blue), are Cy3 and Cy5 fluorochromes in the oligo or FRET also detected, or are the colors reflecting the intensity of Alexafluor fluorescence emission?

Line 318: mRNA => mRNAs

Line 337, legend to Fig 3D: please clarify, what does "aRNA" stand for?

Lines 368 - 372: this is legend to Fig4, and should be moved to line 342.

In the legend to Fig4 B, could you please help the non FRET expert by mentioning the caged hairpin oligo allows FRET from Cy3 (=>fluorescence decrease) to Cy5   

Lines 355-357: please clarify the sentence, there appears to be a confusion between caspases which are proteases (not ribonucleases) and RNA degradation

Author Response

Reviewer 1

The authors are internationally recognized experts in the field of chemical synthesis and use of "caged" oligonucleotides : these are sort of "pro-drugs" to be specifically activated by various stimuli (heat, light, chemical reduction, enzyme activity), adapted to various cell cultures or animal models to be studied.  The present manuscript is an update of a previous review from the same research group published in 2015 (ref 50). 

The manuscript is excellent, very clear and well organized , starting with a general overview of the field and providing more details on the most recent technologies (mostly contributed by the group) i.e. antisense oligonucleotide cyclization, TIVA that allows for mRNA isolation from single cells, and protease-activated oligonucleotide probes.

Comments:

In the introduction could the authors refer to the previous review from their group (ref 50) and to more recent reviews from other groups (such as Vasseur 2018, Chen 2019 ...), and explain the novelty of their contribution in the present manuscript?

We included two recent reviews at the end of the introduction, the ones that we were able to find on this topic. Thank you for this suggestion.

Line 22: please define oligonucleotide as "oligo" here and keep using this abbreviation all over the text (including line 139)

We added definition at the beginning. (Global comment: some line numbers referred to by reviewer 1 are shifted, but we believe we were able to interpret all comments.)

Line 26: could you be more accurate about CRISPR primers and mention small guide RNAs, thus introducing the sgRNA abbreviation that will be used line 110

We changed “primer” to “single-guide RNAs (sgRNAs)”.

Lines 49-50: please clarify/correct English for the legends to Fig1A and B

We added clearer descriptions to the figure.

Line 65: avoid redundancy : mRNA transcripts => mRNAs

We changed as suggested.

Lines 82 and 85: could you be more precise in each case by defining the step at which interference with RISC activity occurs

(We believe this refers to Xinjing Tang’s papers)  Authors of the original work consider that 5′ caging groups will disrupt the formation of RISC complex. More details probably require further study. The context was added as suggested.

Line 88: please clarify/correct English

We changed the phrasing.

Line 100: motif => method, strategy?

This was changed as suggested.

Line 111:  "editing function on GFP" is laboratory slang, confusing protein and DNA molecules! Could you better explain the experimental read out of this study?

We adjusted phrasing.

Lines 112, 126, 127, 151, 153: please standardize numbering in the manuscript, either nucleotides (nt) or bases

This was globally changed to bases.

Line 123: please clarify/correct English

Done

Line 130: please mention it's the reduced form of glutathione, and define GSH that will be used line 137

Changed

(line 154) Changed as suggested

Line 149: => The circle size

Modified to say, “The size of the circular oligo”

Line 163: please clarify "atom-efficient approach"

We added context as suggested.

Line 171: could you add a sentence to explain the mechanism behind light activation of a caged oligonucleotide?

We added context as suggested (line 197).

Line 184 and 198: please standardize NIR => near-IR (similar to near-UV on line 198)

We changed as suggested.

Line 187: define (bpy) here

We changed to bipyridine.

Lines 201-202: avoid redundancy :tunable, fine tuning

We changed as suggested.

Line 204: self cleavable twister ribozyme of Oryza sativa

We changed as suggested.

Line 212: "transfection" refers to cells receiving foreign DNA, e.g. cells are transfected with/by a plasmid, and thus "transfected plasmid" is not correct.

Transfection was replaced with “exogenously supplied”.

Line 305, legend to Fig 2 B: please explain the different colors observed in the figure, resulting from immunofluorescence  (instead of immunohistochemistry as stated) staining of GFAP , and use of a secondary antibody coupled to Alexafluor 647 (based on methods in original publication), DAPI for DNA staining (blue), are Cy3 and Cy5 fluorochromes in the oligo or FRET also detected, or are the colors reflecting the intensity of Alexafluor fluorescence emission?

The signal is only for fluorescence from Alexafluor 647. We added a color bar in the image.

Line 318: mRNA => mRNAs

We changed the one in line 392.

Line 337, legend to Fig 3D: please clarify, what does "aRNA" stand for?

It stands for amplified RNA, this change was added to the caption.

Lines 368 - 372: this is legend to Fig4, and should be moved to line 342.

The position of Figure 4 was adjusted.

In the legend to Fig4 B, could you please help the non FRET expert by mentioning the caged hairpin oligo allows FRET from Cy3 (=>fluorescence decrease) to Cy5   

Language on FRET is added to the 1st example (circular antisense DNA); legend of Fig 4B was also modified as suggested.

Lines 355-357: please clarify the sentence, there appears to be a confusion between caspases which are proteases (not ribonucleases) and RNA degradation

The degradation of RNAs are by ribonucleases; this was changed as suggested. Thank you for catching this error.

Reviewer 2 Report

The manuscript is an interesting short review, summarizing current knowledge about the specifically protected oligonucleotides. I find it beneficial, but I should point out some issues. Especially the organization of the article, resp. the designation and division of chapters do not seem appropriate to me.

Introduction (1.) with several subchapters discussing various aspects including types of oligonucleotides protection and ways of activation is followed by chapters 2. and 3. and 3.(!) discussing the studies of authors dealing again with various/additional types of oligonucleotides protection and way of activation. And further the manuscript ends with Summary numbered as 2.

Additional note concerns the extent of chapters discussing the results of the authors´ laboratory. Since these results were already published, such big extent of their repetition is probably not necessary in a review article.

I think, if the article is intended as a general review, and not a summarization of authors´ published results, the structure should be reconsidered and reorganized. If the latter is the intention, it should be at least renumbered.

The description of Figure 4. is placed incorrectly.

Author Response

Review 2

The manuscript is an interesting short review, summarizing current knowledge about the specifically protected oligonucleotides. I find it beneficial, but I should point out some issues. Especially the organization of the article, resp. the designation and division of chapters do not seem appropriate to me.

Introduction (1.) with several subchapters discussing various aspects including types of oligonucleotides protection and ways of activation is followed by chapters 2. and 3. and 3.(!) discussing the studies of authors dealing again with various/additional types of oligonucleotides protection and way of activation. And further the manuscript ends with Summary numbered as 2.

Additional note concerns the extent of chapters discussing the results of the authors´ laboratory. Since these results were already published, such big extent of their repetition is probably not necessary in a review article.

I think, if the article is intended as a general review, and not a summarization of authors´ published results, the structure should be reconsidered and reorganized. If the latter is the intention, it should be at least renumbered.

We are trying to provide a balanced overview of several general strategies to make caged oligos while also introducing new designs from our laboratory. As for the published results, the goal is to present the novel designs and synthetic improvements, which we believe are worth the length for their context. In the revised version, the contents are in the new sections and renumbered as suggested.

The description of Figure 4. is placed incorrectly.

The position of Figure 4 was adjusted.

Reviewer 3 Report

The review article summarizes the concept and shows some examples of the so-called conditionally activated ("Caged") oligonucleotides. It starts with a general principle of the design of caged oligonucleotide, followed by various activation processes, and ends with three specific contributions from the authors' laboratory. Overall the review is well-written and gives a good balance between the essential information and length.  Its publication is recommended after fixing some minor presentational issues as follows:
- The organization/heading/subheading is unbalanced and rather confusing. Section 1 (introduction) is too large, consisting of too many subheadings, while the rest sections do not have any subheadings. I suggest splitting the introduction into 3 sections 1. introduction 2. methods for blocking the function of oligonucleotides 3. Activation stimuli 4. Specific examples from the authors' laboratory (circular light-activated DNA, TIVA, and caspase-activated probe) 5. Summary
- The caption of Figure 4 was in the wrong place.
- The section summary begins with the history of the term "caged"... This should better move to the introduction. It is a kind of too late to put it here.
- All previously published figures, either being used as such or with adaptation need specific copyright permission from the original publisher.

Author Response

Review 3

The review article summarizes the concept and shows some examples of the so-called conditionally activated ("Caged") oligonucleotides. It starts with a general principle of the design of caged oligonucleotide, followed by various activation processes, and ends with three specific contributions from the authors' laboratory. Overall the review is well-written and gives a good balance between the essential information and length.  Its publication is recommended after fixing some minor presentational issues as follows:
- The organization/heading/subheading is unbalanced and rather confusing. Section 1 (introduction) is too large, consisting of too many subheadings, while the rest sections do not have any subheadings. I suggest splitting the introduction into 3 sections 1. introduction 2. methods for blocking the function of oligonucleotides 3. Activation stimuli 4. Specific examples from the authors' laboratory (circular light-activated DNA, TIVA, and caspase-activated probe) 5. Summary

Contents were put into these new sections as suggested. Thank you for this format.

- The caption of Figure 4 was in the wrong place.

The position of Figure 4 was adjusted.

- The section summary begins with the history of the term "caged"... This should better move to the introduction. It is a kind of too late to put it here.

We introduced the concept in the introduction section (line 36). We think it’s useful stylistically to include additional background in the summary, to tie together the whole story.

- All previously published figures, either being used as such or with adaptation need specific copyright permission from the original publisher.

Permissions from the publishers are included.